# Immigrants resettlement in developing countries: A data-driven decision tool applied to the case of Venezuelan immigrants in Colombia

**Gina Galindo** [1]*, **Jose Navarro**[1], **Jhonattan Reales**[1], **Jhoan Castro**[1], **Daniel Romero**[1], **Sandra Rodriguez A.**[2], **Daniel Rivera-Royero** [1]

**1** Department of Industrial Engineering, Universidad del Norte, Barranquilla, Colombia, **2** Department of Economics, Universidad del Norte, Barranquilla, Colombia

\* ggalindo@uninorte.edu.co

## Abstract

Immigrants' choice of settlement in a new country can play a fundamental role in their socio-economic integration. This is especially relevant if there are important gaps among these locations in terms of significant factors such as job opportunities, quality of health service, among others. This research presents a methodology to perform a recommended geographic redistribution of immigrants to improve their chances of socio-economic integration. The proposed methodology adapts a data-driven algorithm developed by the Immigration Policy Lab at Stanford University to allocate immigrants based on a socio-economic integration outcome across available locations. We extend their approach to study the immigration process between two developing countries. Specifically, we focus on the case of the arrival of immigrants from Venezuela to Colombia. We consider the absorptive capacity of locations in Colombia and include the health and education needs of immigrants in our analysis. From the application in the Venezuelan-Colombian context, we find that the proposed redistribution increases the probability that immigrants access formal employment by more than 50%. Furthermore, we identify variables associated with immigrants' formal employment and discuss specific strategies to improve the probability of success of vulnerable immigrants.

## 1. Introduction

In general, when public policies seek to organize the immigrant settlement process, the objective is to maximize their chances of success. One of the issues that have attracted particular interest among academics in recent years is the construction of analytical frameworks that relate the preferences, characteristics and skills of immigrants to the needs and capabilities of the host country's localities [1–7]. These approaches are mainly based on a two-sided allocation system, which considers: (1) the immigrant individuals or families, and (2) the host locations where they are to be allocated. In these approaches, the immigrant is suggested a location

**Funding:** This work was funded under Prime Cooperative Agreement Number AID-OAA-A-11-00012 entered into by and between the National Academy of Sciences and the United States Agency for International Development (USAID) through the program Partnership for Enhanced Engagement in Research (PEER) Cycle 8. For further information, visit https://sites.nationalacademies.org/PGA/PEER/index.htm. The funders had no role in study design, data collection and analysis, decision to publish, or preparation of the manuscript.

**Competing interests:** The authors have declared that no competing interests exist.

where he or she would have a higher probability of success in order to optimize a predefined outcome [6]. One of the most relevant articles in this regard is [5], which presents a data-driven algorithm for matching locations with household cases in order to maximize the total economic integration outcome of immigrant families.

Research on the migration process is highly relevant and pertinent, as it is estimated that, by 2019, more than 272 million people lived in foreign countries [8]. In particular, there is a need for economic integration policies in developing countries, as pointed out by [9], since 44% of the world's immigrants in 2019 were concentrated in less-developed regions, which includes all regions of Africa, Asia (excluding Japan), Melanesia, Micronesia and Polynesia and Latin America and the Caribbean. In fact, intra-regional migration in South America has been growing at a faster rate than emigration to more developed countries [10]. Among the recent migration waves, the one caused by the economic crisis in Venezuela stands out due to its magnitude [11]. In this regard, more than 3 million people have emigrated from Venezuela, among which approximately 1,800,000 have settled in Colombia [12].

In this paper, we discuss a methodology for the identification of potential resettlement locations that can be recommended to immigrants in a developing country, along with an application to the specific case of immigrants coming from Venezuela to Colombia. The recommended location seeks to optimize the economic integration and welfare outcomes of immigrants. Similar to the algorithm given in [5], in this research the algorithm looks into the historical success of the economic integration of immigrants in each location and relates that success to key characteristics of immigrants, (e.g. age, occupation). It then predicts a probability of economic integration and welfare outcome for each immigrant in each location. And finally, the algorithm matches locations with household cases to maximize the total economic integration outcome of immigrant households, while respecting the constraints related to the maximum number of immigrants per region. Adapting the algorithm presented in [5] to a developing country context, while addressing household needs related to education and health, imposes important challenges that are addressed in this research.

The proposed approach should help to balance the distribution of immigrants across locations in Colombia. In this regard, Venezuelan immigrants in Colombia have naturally settled in both border regions and major cities. The capital, Bogota, has received about 20% of Venezuelan immigrants; La Guajira, which is one of the poorest regions, has received approximately 8,5%. One of the consequences of this situation is that some regions have been overwhelmed by a large number of immigrants, which have exceeded the resources of the localities.

In the literature, when analyzing the integration process of immigrants, the most relevant aspects considered are related to (1) physical and emotional health, (2) social interactions, and (3) economic conditions. The research in [13] suggests 10 key areas of integration: employment, housing, education, health, social bridges, social bonds, social links, language and cultural knowledge, security and stability, and rights and citizenship. An important remark is that these aspects are interrelated. For instance, according to [14], the subjective economic situation of immigrants is associated with their psychological well-being. Furthermore, [15] points out that a low level of education is associated with poor economic integration, and that through formal education, it is possible to rapidly increase immigrants' income, thus also improving their standard of living. In [1], the authors also stress that future research should pay more attention to other objectives such as health or education, in addition to economic or social integration. On the other hand, there is a small body of literature on allocation models that focuses on vulnerable populations or those with specific health or education needs (e.g. the elderly). One study that can be mentioned in this regard is the research of [5], which considers the possible health needs that an individual may have. Some variants of this type of

approach that also consider the preferences and needs of immigrants without neglecting their success of their economic integration are given in [1, 16], and [17].

There is also a need to develop tools to allow immigrants to have better information about the particular conditions of the locations in the host country. In this regard, the prominence of certain locations internationally may bias the selection of immigrants and make them want to be assigned to certain locations just because they are the best known. However, these places of settlement may play a key role in the integration process of immigrants. As [9] points out, immigrants who are assigned to less favorable locations perform worse in terms of employment and education compared to immigrants assigned to more prosperous areas. Therefore, it is important to provide immigrants with objective information about all locations, so that they can better establish their preferences [17].

To the best of our knowledge, our research is the first to address the issue of socio-economic integration of immigrants arriving in a developing country, to suggest a geographic redistribution of immigrants to more beneficial locations, based on their demographic characteristics and historical data. Our research adapts the model proposed in [5] and expands the analysis to obtain insights specifically focused on vulnerable populations, such as the elderly. Our methodology focuses on the well-being of immigrants and includes health and education needs from the immigrants' point of view, along with the perceived capacity of candidate locations to adequately host immigrants. The analysis of location capacity is necessary because in developing countries, the standard of living and the infrastructure among cities are not homogeneous and there are significant gaps in health and education coverage [18, 19]. Our results can be used to deliver better information to immigrants about their expected integration in potential settlements.

## 2. Methodology

The methodology to develop the predictive and allocation models is based on [5], which includes the following stages: (1) data designation, (2) modeling, (3) mapping, and (4) matching. Below, we briefly describe each of these stages and how we have adapted them in this research. We refer the reader to [5] for more details regarding these stages.

### 2.1 Data designation

This first stage designates the data set to be used in the analysis, which is given by individual immigrants, along with their full set of covariates. For the subset of data used for model training, we also need their integration outcome. The databases used are publicly available from the Open Science Framework database, DOI 10.17605/OSF.IO/4YRJN.

### 2.2. Modeling

In this state, several statistical models are built and tested to predict the probability of the integration outcome for each immigrant in the available locations. For the applications in [5], the integration outcome is defined as the probability of finding a job. The goal of this stage is to fit a predictive model for each location. The result is a matrix M that gives the predicted probabilities for each immigrant at each location.

For our approach, we have used immigrants' location from historic data as a covariate instead of fitting a different model for each location. This allows us to compare probabilities that are estimated using a unique predictive model for all of the locations. The algorithms tested in this stage are Random Forest, Extreme Gradient Boosting (XGBoost) and Support Vector Machine (SVM). These algorithms were selected since they are well-known to have a satisfactory performance in classification problems. Random forest and XGBoost are ensemble

models based on multiple classification trees for higher performance. The SVM algorithm works based on the concept of hyperplanes to define regions to classify the observations in the different classes being considered. For testing purposes, the set of observations was divided into training and test sets, with a ratio of 85/15. Then the training set is also divided into training and validation sets, with a ratio of 90/10. In addition, a 10-fold cross-validation repeated 4 times is used to avoid overfitting of the model. The algorithms are compared based on statistical metrics. Once the best classification model is selected, it is required to check whether the probabilities are calibrated. This implies assessing how accurately the estimates compare to the true probability of an observation belonging to the predicted class. At this stage we implement the Platt scale method to calibrate the output of the classification models to enhance predictions. The models were built using caret package version 6.0–86 in R version 4.0.2.

In order to better adapt the methodology to developing countries, the integration outcome has been defined as the probability of finding formal employment. The importance of formal employment lies in the fact that it guarantees access to social security and health services, facilitates access to legal labor rights and reduces exposure to labor exploitation, which is not uncommon in developing countries.

## 2.3. Mapping

In the mapping stage, individuals are aggregated into household cases, following a mapping function. In the application presented in [5] to the US and Switzerland, the mapping function is defined as the predicted probability that at least one household member finds a job in the given location. From this stage, the matrix $M^*$, which contains the case-location metric $\gamma_{ij}$ of the household $i$ in the location $j$, is obtained. The importance of the mapping stage is that it keeps households together during voluntary redistribution. An assumption of the methodology is that families do not want to be split across multiple locations.

The mapping function defined in this research is given by the probability that at least one of the members obtains formal employment for households with less than 5 members; for households with 5 or more members, the metric is the probability that at least two or more members obtain formal employment.

## 2.4. Matching

In this last stage, migrant households are suggested a location in the host country, based on a mathematical model that optimizes a given criterion related to the chosen case-location metric, subject to a set of constraints. In [5] there is a given number of slots, $t_j$, which limits the number of household assignments to each location. In addition, it is possible to add constraints so that households with special needs (e.g. severe medical conditions) are not assigned to certain locations. The objective function minimizes the sum total cost, $c_{ij}$, defined as the cost of household $i$ for each location $j$, which is calculated as $c_{ij} = 1 - \gamma_{ij}$.

In developing countries, such as Colombia, information on the parameter $t_j$ may not be available. We propose to estimate $t_j$ based on an absorptive analysis per location, combined with data on (1) the percentage of migrants over the country's total population, (2) the percentage of the country's total population residing in each location, and (3) the percentage of immigrants relative to each location. Absorptive capacity analysis assesses the conditions of the locations to receive and integrate migrants based on multiple socio-economic parameters. The first step is to identify appropriate metrics, which should include economic, welfare, employment, health and educational aspects. From these socio-economic metrics, a clustering method is proposed to find groups of locations with similar outcomes. A k-means algorithm from the Stats package in R is used. This algorithm identifies k groups of observations.

Locations with equivalent metrics will be in the same group and separated from other groups with different characteristics. The objective is to identify the locations with better and worse conditions to provide an adequate level of quality to their inhabitants. This information is used in the allocation phase to define the capacities of the locations.

In our optimization model, in addition to considering the cost $c_{ij}$, we include penalties for locations with poor performance in two dimensions: health and education. For health, we evaluate performance using metrics on health coverage and uninsured health rate. For education, the metrics are illiteracy rate and low educational attainment. Locations with poor indicators in these metrics are assigned a penalty that is applied to households with special needs in terms of priority health care (e.g. of infants and the elderly), or educational services (e.g. children under 13 years old). Locations with a health coverage index under the 20th percentile (83.2%) are penalized. Locations with an uninsured health index above the 80th percentile (14.3%) are penalized too. Locations above the 80th percentile in illiteracy rate and educational achievement are also penalized (18.7% and 62.9% respectively). As a result of this analysis, we have two matrices of penalties: (1) matrix S for health requirements and attention to vulnerable people, and (2) matrix E, for penalties related to educational issues.

Additional penalties are assigned in cases where a relocation improves household metric below a given threshold. The relocation threshold reduces the number of assignments when the economic integration is not substantial. This parameter can be adjusted by decision makers, where small values will favor a large number of relocations and large values will assign new locations to households when the improvement in probability is notable. In addition to sensitivity analysis another possible method for estimating the relocation threshold is to estimate the minimum probability difference to consider the change in location to be significant using inferential statistics.

The allocation problem is formulated as a mixed-integer programming model where the decision variables assign the families to one of the available locations considering the $t_j$ indices, as well as the S and E matrices. The formulation of the model is as follows:

$$x* = argmin_x f(x), \tag{1}$$

where

$$f(x) = \sum_{i=1}^{m} \sum_{j=1}^{n} \left( \alpha c_{ij} + \beta S_{ij} + \delta E_{ij} \right) x_{ij} \tag{2}$$

Subject to the following restrictions:

$$\sum_{i=1}^{m} x_{ij} \leq t_j, \ \forall j \tag{3}$$

$$\sum_{j=1}^{n} x_{ij} = 1, \ \forall i \tag{4}$$

$$x_{ij} \in [0, \ 1], \ \forall i, j \tag{5}$$

The objective function is given by (1), where $\alpha$, $\beta$ and $\delta$ are the weights of the cost, health care penalty and education penalty, respectively. Constraints (3) guarantee that the number of households assigned to each location should not exceed its suggested maximum capacity. Constraints (4) allocate each household to a single location. Constraints (5) ensure that decision variables are binary, where $x_{ij}$ equals 1 if household $i$ is assigned to location $j$. The solution of the matching model given by (1) to (5) provides the suggested location to the immigrant households.

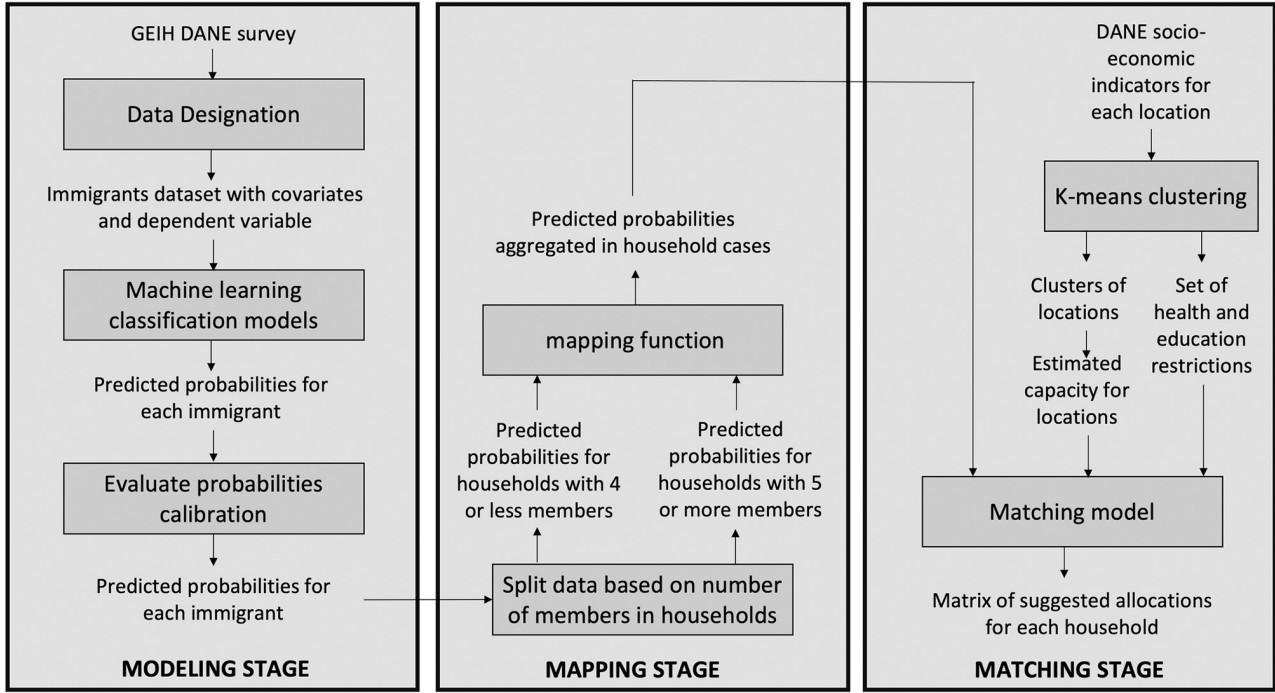

**Fig 1. Methodology of the algorithm.**

Fig 1 summarizes the methodology implemented in this study. The following section discusses the application of the proposed methodology to the phenomenon of immigrants arriving to Colombia from Venezuela.

## 3. Results for Colombia

For the application of the proposed methodology to data on immigrants from Venezuela in Colombia, the database used comes from the Large Household Survey in Colombia (GEIH), which is a survey applied monthly by the National Administrative Department of Statistics in Colombia (DANE). GEIH collects information on socio-economic variables (e.g. employment, sex, age, income, household information) of people living in Colombia. There is also a special migration module to collect data on immigrants in Colombia. The data used in this research corresponds to the time period from January 2019 to March 2020. We consider the following variables:

- **Household**: person or group of persons, related or not, who occupy all or part of a home; they cover basic needs with a joint budget and generally share meals.

- **Locations (Departaments)**: correspond to 24 geographic areas in Colombia.

- **Gender**: defined as a two-level categorical variable indicating sex (male or female).

- **Age**: defined in 5 ranges by DANE.

- **Kinship**: status of the person in the household (e.g. household head).

- **Marital status**: marital status of the respondent.

- **Educational level**: the highest level of education attained by the respondent.

- **Previous work**: two-level variable that indicates whether or not the person has had previous work experience.

- **Occupation**: levels defined according to the DANE criteria of the International Standard Industrial Classification of all economic activities (ISIC). This variable contains 55 levels

- **Time in Colombia**: the time in years that the respondent has been in Colombia since entering the country.

- **Available working hours**: the number of hours the respondent is willing to work per week.

- **Occupation status**: indicates whether the respondent is formally employed, informally employed, self-employed/independent, or unemployed. This variable is used as the response variable for our predictive models.

The levels of the *Occupation* and *Location* variables are shown in S1 Appendix and S1 Table, respectively. The levels of the remaining variables are presented in Table 1. For our analysis, we only consider the working-age population (i.e. from 13 years old, according to DANE). Our dataset is composed of 21,132 immigrants from Venezuela, which belong to 11,044 households, with an average of 1.9 persons per household. An important column in the DANE dataset is the expansion factor, which indicates that each record in the data is actually representative of a larger group of individuals with the same profile. Table 1 shows a summary of how the observations are distributed within the levels of the most relevant variables. Also, the population of each location is shown in S1 Table.

Table 2 shows a summary of the performance metrics of the three tested algorithms at the modeling stage, while Fig 2a–2c show the calibration plots of Random Forest, XGBoost and SVM, respectively. The input variables are those specified in Section 2. Notice that for all three algorithms, the probabilities without calibration give better performance. As can be seen, SVM shows less favorable performance metrics than the other two algorithms. The SVM calibration plot also shows more worrisome differences between predicted and observed probabilities. On

**Table 1. Summary of variables and their levels.**

| Variable | Total | Percentage | Variable | Total | Percentage |
|---|---|---|---|---|---|
| **Gender** | | | **Educational level** | | |
| Male | 11,653 | 0.551 | Kinder | 2 | 0.000 |
| Female | 9,479 | 0.449 | College or postgraduate education | 5,160 | 0.244 |
| **Age** | | | Complete high school | 8,758 | 0.414 |
| (18,26] | 6,130 | 0.290 | Incomplete high school | 4,700 | 0.222 |
| (26,59] | 13,259 | 0.627 | Primary education | 2,274 | 0.108 |
| (14,18] | 1,005 | 0.048 | No education/ Unknown | 238 | 0.011 |
| (59,100] | 662 | 0.031 | **Previous Work** | | |
| (11,14] | 76 | 0.004 | Yes | 18,312 | 0.867 |
| **Kinship** | | | No | 2,820 | 0.133 |
| Household head | 7,842 | 0.371 | **Time in Colombia** | | |
| Another relative | 4,030 | 0.191 | Less than 1 year | 6,236 | 0.295 |
| Family group member | 6,809 | 0.322 | Between 1 and 5 years | 13,970 | 0.661 |
| Another non-relative | 2,451 | 0.116 | More than 5 years | 926 | 0.044 |
| **Marital status** | | | **Available working hours** | | |
| Single | 4,994 | 0.236 | (0,39] | 1,988 | 0.094 |
| Married/common-law relationship | 12,672 | 0.600 | (39,48] | 9,647 | 0.457 |
| Separated/ Widow(er) | 3,466 | 0.164 | (48, 130] | 9,497 | 0.449 |

**Table 2. Machine learning algorithms' performance.**

| Model | Area under the ROC curve (AUC) | Balanced accuracy | Sensitivity | Specificity | F-Measure | Log-Loss with calibration | Log-Loss without calibration |
|---|---|---|---|---|---|---|---|
| **Random Forest** | 0,7676 | 0,6956 | 0,6919 | 0,6993 | 0,4776 | 0,462 | 0,426 |
| **XGBoost** | 0,7824 | 0,7173 | 0,7662 | 0,6685 | 0,4952 | 0,413 | 0,424 |
| **SVM** | 0,7396 | 0,6881 | 0,6840 | 0,6922 | 0,4689 | 0,470 | 0,469 |

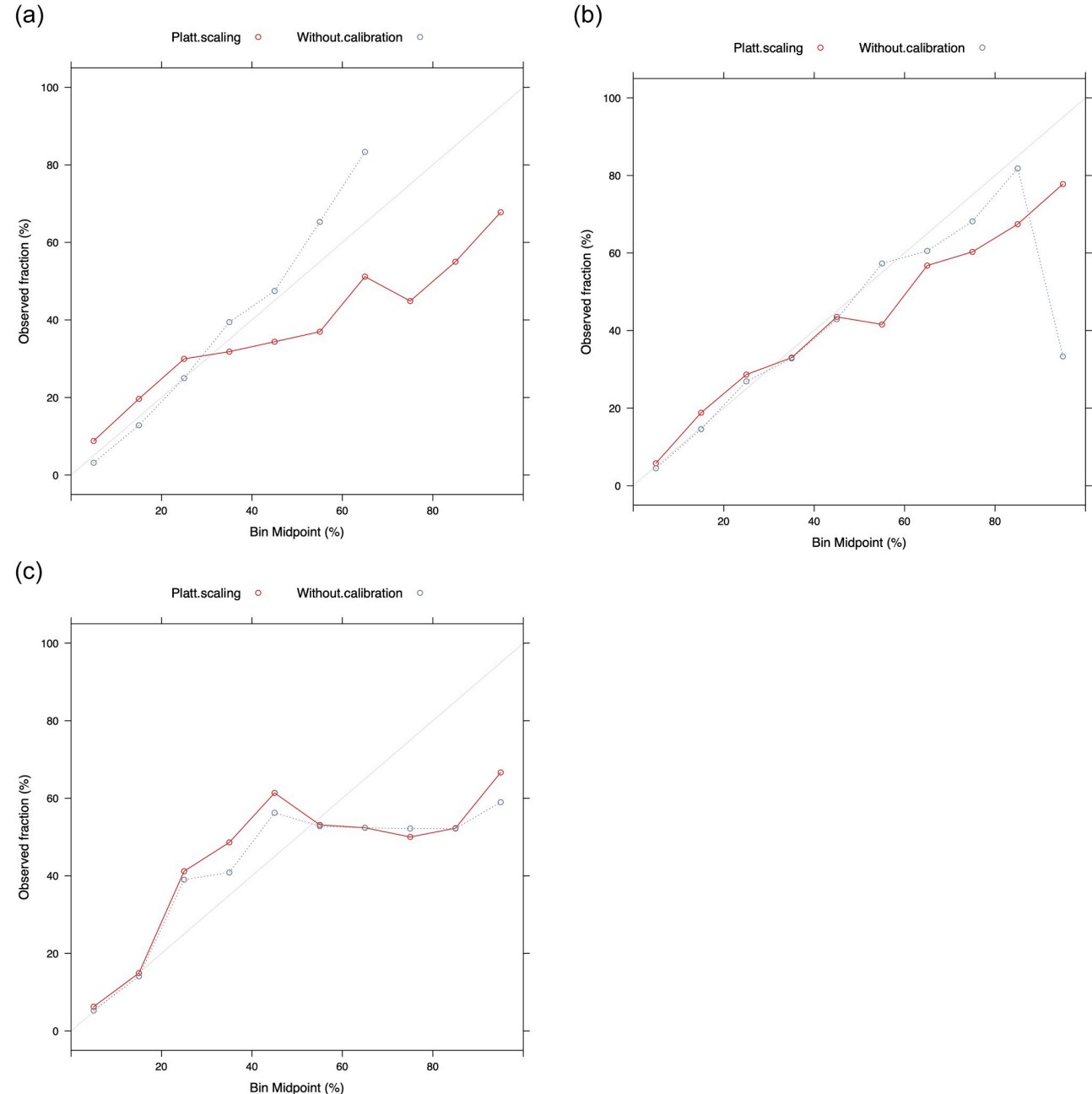

**Fig 2.** a. Calibration plot for Random Forest. b. Calibration plot for XGBoost. c. Calibration plot for SVM.

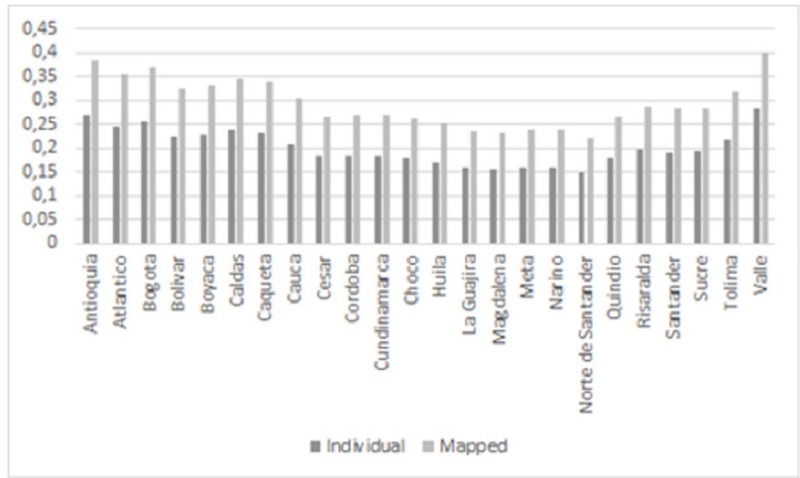

**Fig 3. Predicted probabilities for individuals and households by locations.**

the other hand, XGBoost and Random Forest show similar results in terms of performance metrics and calibration graphs.

We proceeded to a validation of the results of XGBoost and Random Forest against the real characteristics of the locations, e.g. the level of employment, the economic indicator of poverty. In this regard, the analysis provides different perspectives on both models that allowed us to select Random Forest over XGBoost. In particular, XGBoost provides the highest predictive probabilities of formal employment in the locations with the lowest economic indicator of poverty and employment in Colombia. One of the reasons for this behavior is that XGBoost tends to provide extreme probabilities (close to 0 and 1) for certain types of immigrant profiles, especially those that are less common in the sample. Random Forest, on the other hand, avoids high employment probabilities in locations with habitually low probabilities of formal employment and high poverty indicators. According to this last step, although XGBoost and Random Forest have a competitive performance, Random Forest provides results more consistent with the Colombian reality. Therefore, the Random Forest algorithm is selected for the next stages of the study. In addition, based on the calibration results, we have decided to use the uncalibrated versions of the probabilities.

Fig 3 shows the mean prediction probabilities generated by the Random Forest algorithm. The probabilities for individuals in locations such as Valle Del Cauca, Antioquia and Bogotá are the highest. In contrast, Norte de Santander, Magdalena and La Guajira do not show good employment prospects. Mapped probabilities tend to be higher than probabilities for individuals. Moreover, the mapped probabilities are more similar across locations than the individual probabilities. An important remark is that the mean predicted probabilities of finding formal employment are less than 0.4 in all cases. S2 Table provides more details on the predicted probabilities.

Based on an analysis of the available socio-economic indicators most suitable for the absorption analysis, we identify a set of 28 indicators, which are summarized in Table 3. More information about these indicators can be found in S3 Table. Next, the k-means clustering method results in the clusters shown in Fig 4, numbered from left to right as Cluster 1 to Cluster 4. The socio-economic metrics are best in the descending order of the clusters (i.e. the best metrics are found for cluster 1, while the worst values correspond to cluster 4).

Currently, the average percentage of immigrants in Cluster 1 locations is 2.75%; 2.38% in Cluster 2; 3.34% in Cluster 3; and 7.16% in Cluster 4, where the latter one comprises the lowest

**Table 3. Metrics for assessing the locations' socio-economic conditions.**

| Well-being | Employability | Health | Development | Economy | Education |
|---|---|---|---|---|---|
| Housing deficit | Ratio between job offers and demand | Subsidized insurance scheme | Human Development Index (HDI) | Gross domestic product (GDP) | Low level of education |
| No access to improved water sources | Rate of informal employment | Contributory insurance scheme | Multi-dimensional Poverty Index (MPI) | Per capita GDP at current prices | Illiteracy |
| Inadequate housing material | Long-term unemployment rate | Barriers to early childhood care services | MPI intensity | Monetary poverty | Backwardness in school |
| Inadequate disposal of excreta | Child labor | Uninsured population | Population | Extreme poverty | School absenteeism rate |
| Critical overcrowding rate | Unemployment rate | Barriers to access to health services | Percentage of immigrant population | | |

performing locations in the socio-economic metrics of our analysis. We set a limit for the number of families to be assigned to each location, in order to assign more families to clusters with better socio-economic metrics. For cluster 1, we defined that at most 6% of its population would be immigrant; 4% for Cluster 2; 2.5% for Cluster 3; and 1% for Cluster 4. These percentages are close to those provided by the model when no upper limit is set for the locations.

Fig 5 shows the redistribution of the immigrant population after applying our proposed approach. The direction of the flow is from left to right. As can be seen, there are three main locations where the algorithm concentrates the allocation of immigrant families, which are Valle, Bogotá and Antioquia, which belong to Cluster 1. On the other hand, the algorithm recommends a significant reduction of the immigrant population in several locations, among which Norte de Santander, La Guajira, Magdalena and Cesar stand out. This is especially relevant since Norte de Santander and La Guajira are border regions that generally attract the arrival of immigrants. It is worth noting that these locations belong to clusters with less favorable socio-economic metrics. An important remark is that the algorithm leaves more than 33.5% in their current location.

## 4. Discussion and insights

In this section we highlight some important insights based on the results of the prediction model and the redistribution recommended by the data-driven algorithm. We attempt to

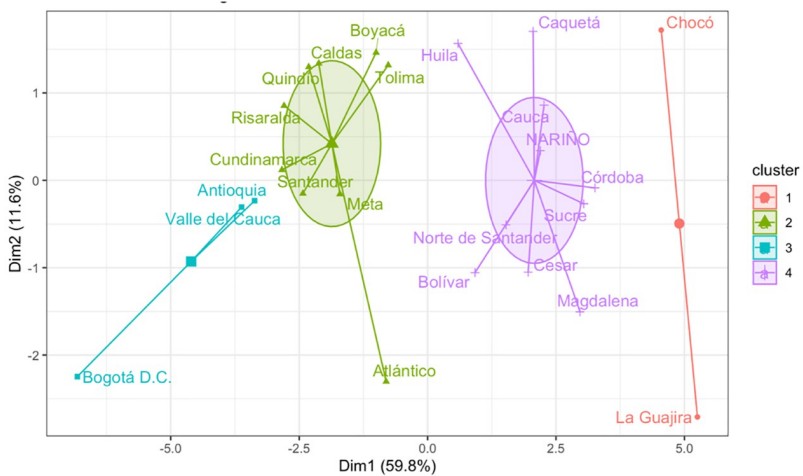

**Fig 4. Results from the cluster analysis.**

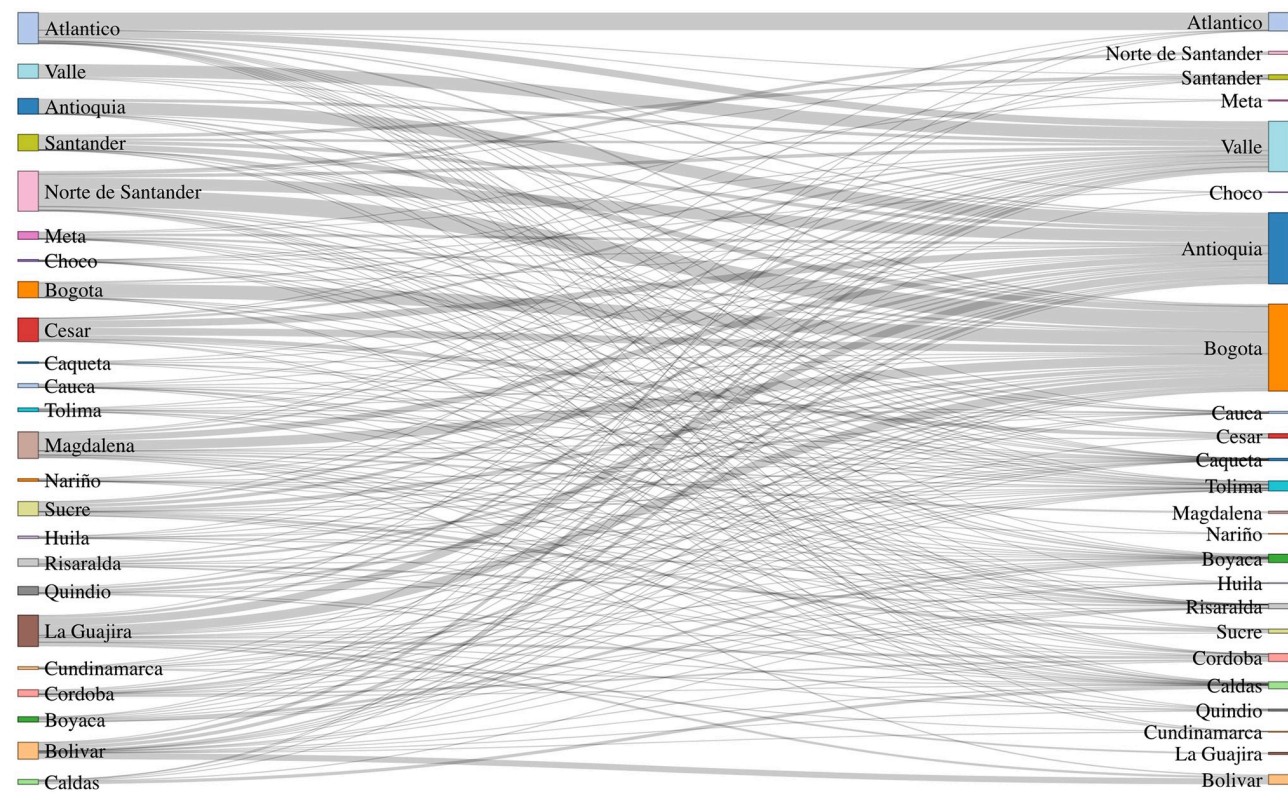

**Fig 5. Recommended redistribution of immigrant households.**

answer the following questions: (1) How does the proposed algorithm improve the chances of finding a formal job? (2) What is the impact of the penalties associated with health and education included in our matching model on the allocation delivered by the algorithm? (3) What characteristics are most and least favorable for immigrants to obtain formal employment? (4) Which subgroups of the immigrant population show more difficulties in finding formal employment?

Analyzing the distribution recommended by the algorithm, we find that the mean value of the integration outcome for households (as defined in Section 4.2) improves by 57.2% (0.107 in absolute values) compared to the current distribution. Fig 6a shows the histogram of the metric with the current distribution; Fig 6b, corresponds to the recommended redistribution; while Fig 6c shows the result of the difference between the metric in Fig 6a and 6b. Notice that in the proposed redistribution, the metrics are distributed approximately uniformly around the mean, whereas in the current distribution, the histogram is skewed to the right.

From the recommended redistribution information shown in Fig 5, it is possible to see trends with respect to some of the potential migration flows. For instance, a large percentage of households in Norte de Santander would be recommended to move to Bogotá. Most households from La Guajira would also be recommended to move to Bogotá and Antioquia, and a minority would be spread across the Caribbean region of Colombia. It would be valuable to use this information when designing programs to advise and support migrant resettlement.

Out of the 11,044 households assessed, 8.8% have on average probabilities of formal employability below 15% for all locations assessed. These households are in a vulnerable situation since they are below percentile 10 in terms of probability. The algorithm suggests leaving

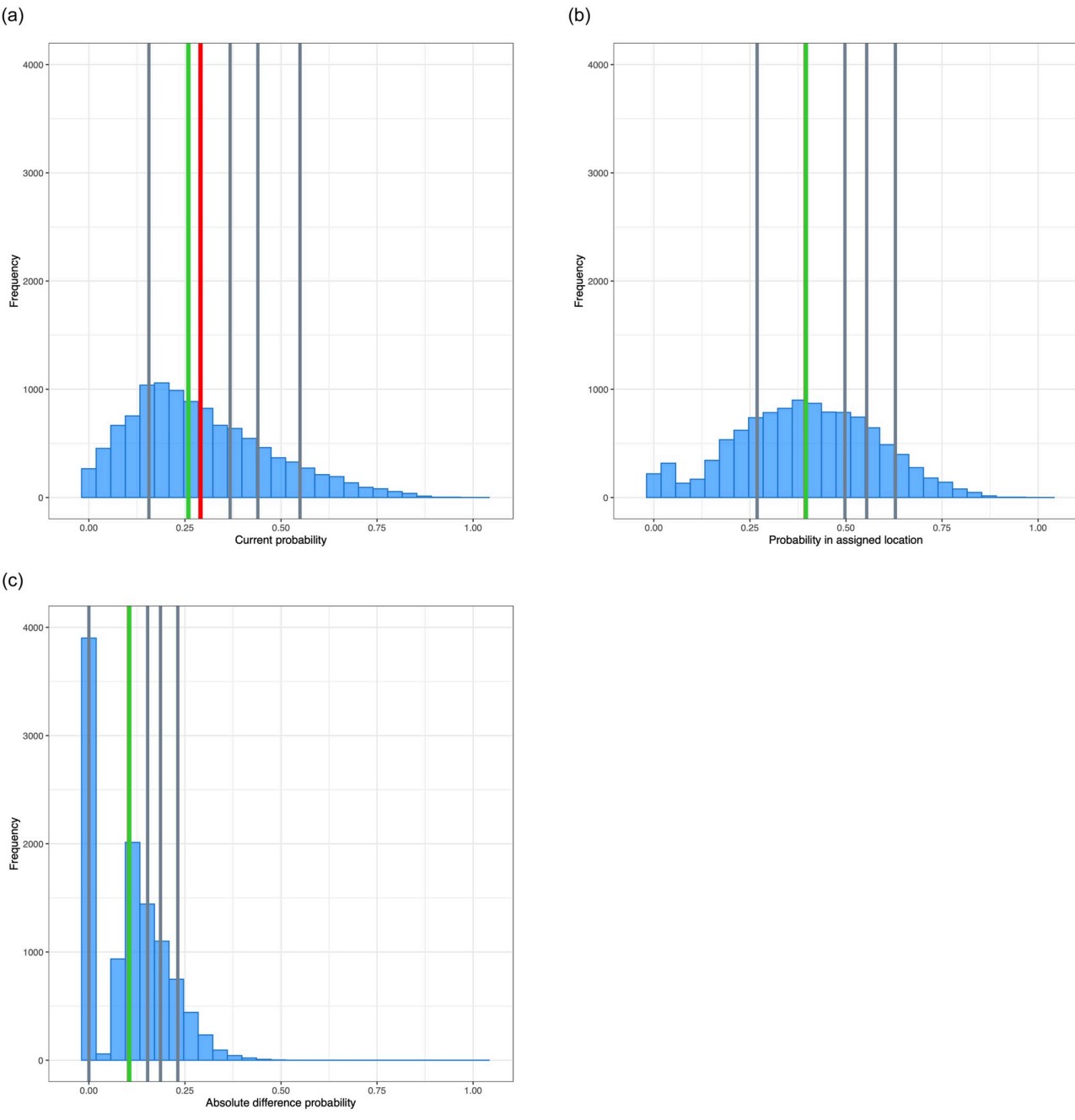

**Fig 6.** a. Metrics for households under current geographic distribution. b. Metrics for households under the recommended geographic distribution. c. Difference between the metrics under the current and the recommended geographic distribution.

66.5% of these households in their current location. For this subgroup, the average improvement in the probability of formal employability is 75.45% (0.026 in absolute terms).

To assess the impact of education and health penalties, we compare results with and without such penalties. Locations with health and education penalties are assigned 46 households with special needs when the penalties are active versus 89 households when they are deactivated. In locations that belong to Cluster 1, the mean improvements in terms of probability of finding formal employment are slightly reduced when penalties are included. It appears that

with penalties, some households with special needs are assigned to locations where they are less likely to find formal employment as long as they avoid the penalized locations. For example, the recommended redistribution assigns 4.65% fewer households to Bogota when the penalties are active, while the improvement in the average probability of formal employment for households with the penalties is 73.5% versus 71.5% without them. Although this behavior is observed for Cluster 1, the overall average probability of finding formal employment remains the same whether penalties are included or not.

We conducted a sensitivity analysis to assess the impact of education on the socio-economic outcome of households. In this sense, the largest impact occurs for the subgroup of immigrants with an educational level below the high school category (e.g. elementary school). If the educational level for these immigrants is changed to "completed secondary", 83.7% of households improve their socio-economic outcome, with a resulting average improvement of 6%. Education-related efforts could focus on helping immigrants to achieve at least a full secondary level of education.

From the modeling stage of the methodology, it is possible to identify the most significant variables that affect the probability of formal employment of individuals, which are (1) gender, (2) work experience, (3) time since their arrival in Colombia, (4) educational level, and (5) occupation. Table 4 shows the most and least favorable characteristics of immigrants in relation to obtaining formal employment.

From Table 4, domestic work stands out as an occupation with lower probabilities of having a formal job. But this is not a peculiarity for the immigrant population; traditionally in Colombia, it is common for families to hire informal household help. Other occupations with a relatively lower probability of formal employment in Colombia are private domestic services, other service activities, and land transportation. All of these occupations are traditionally part of the informal economy in Colombia. Efforts to improve labor conditions in these fields would benefit migrants and the Colombian community. On the other hand, the occupational fields with the highest frequency of individuals with formal jobs are real estate, education, social and health services, sports and cultural activities and clothing manufacturing. Analysis on the occupation fields less and more favorable to immigrant integration would be even more valuable if adjusted to the immigrants' skills and the labor market of each location. Information in this regard can be useful when designing training and education programs.

According to our results, older immigrants and women tend to have lower predicted probabilities of finding formal employment. As for the older immigrant group (over 59 years old), if we take into account the redistribution suggested by the model, the percentage improvement in the probability of finding formal employment for this group is 59.5%, which represents 0.093 in absolute values. For this subgroup, their average probabilities of obtaining formal employment are relatively low (0.152 on average) compared to those found for the general immigrant population, with values ranging between 0.173 and 0.297 on average. For this subgroup, the magnitude of the average improvement in the probability of finding a formal job, achieved with the

**Table 4. Most and least favorable immigrant profiles for obtaining formal employment in Colombia.**

| Characteristic | Most favorable | Least favorable |
|---|---|---|
| Gender | Male | Female |
| Work Experience | Yes | No |
| Age | 18–26 | over 50 |
| Educational level | Undergraduate/graduate | None |
| Occupation | Real estate; education; social services; cultural activities | Domestic work |
| Time since their arrival | More than 5 years | Less than 1 year |
| Available working hours | 39–48 | less than 39 |

redistribution, is inverse to the corresponding education level. In this sense, for older adults with no education at all, the average improvement is 76%; primary school, 64.5%; incomplete secondary school, 67.5%; complete secondary school, 51.9%; and higher education, 31.7%.

As for the female population, in the current geographic distribution, their average probability of formal employment is 16.2%, while for men is 23.1%. The proposed redistribution generates an improvement in the probability of finding formal employment of 56.2%, which is similar to that observed for the male population (57.5%). Initially, 50% of the women were concentrated in the departments of Norte de Santander, Atlántico, La Guajira, Magdalena and Cesar, where the average probability of finding formal employment is 12.1%, 19%, 10.9%, 10.9%, and 13.6% respectively. The allocation model suggests a redistribution such that the five departments with the highest concentration of women are Antioquia, Bogota, Valle, Atlántico and Tolima. In these departments, the average probability of finding formal work for women would be 22.6%, 22.3%, 26%, 22.7% and 24.4% respectively.

## 5. Conclusions

In this research we have presented a methodology for identifying a geographic redistribution of immigrants arriving to a developing country with high rates of informal employment, limitations in its capacity to absorb the incoming immigrant population, and important gaps in terms of socio-economic indicators among potential resettlement sites. We explore the specific case of Venezuelan immigrants in Colombia, in order to improve their chances of socio-economic integration. The proposed methodology focuses on maximizing the probability of obtaining formal employment, while considering the relevant social needs to be met. The results of the application of the proposed methodology to a representative sample of immigrants show that, in general, the average probability of obtaining formal employment in Colombia for Venezuelan immigrants is below 0.3 in the different locations in Colombia. According to our results, there are certain immigrant profiles and certain occupations that increase the probability of finding formal employment. In general, young male immigrants with undergraduate or graduate studies tend to be more likely to find formal employment. On the other hand, women and older adults tend to show lower probabilities. An interesting finding indicates that some locations in Colombia where immigrants are currently concentrated are not the most favorable for improving their chances of finding formal employment. The results from the proposed methodology can be used by immigrants to make more informed decisions about their settlement destination in Colombia. The results also provide relevant information for authorities and organizations that can contribute to the design of future policies and programs for the socio-economic integration of immigrants in Colombia.

As future work, it is important to be able to dynamically update the recommended distribution in order to incorporate the changing environment of the migration process. In this sense, the generation of new, updated data is of great value. Future analyses could include household monetary income as a response variable, which combined with the probability of obtaining formal employment, will provide valuable insights. Given the current context, a relevant analysis would be to assess how COVID-19 has impacted the socio-economic integration of immigrants. Finally, it would be valuable to explore new ways to include immigrants' preferences and model cultural similarities with host communities to further boost their socio-economic integration.

## Supporting information

**S1 Appendix. Levels of the variable corresponding to occupational fields.**
(DOCX)

**S1 Table. Distribution of immigrants in Colombian locations.**
(DOCX)

**S2 Table. Predictive probabilities with the random forest model.**
(DOCX)

**S3 Table. Descriptions of the socio-economic indicators for the absorptive analysis.**
(DOCX)

## Acknowledgments

We would like to specially thank Dr. Duncan Lawrence, Dr. Jens Hainmueller, and Jennifer Fei from the Immigration Policy Lab for accompanying us in this research and for providing their support and feedback. We would also like to thank Gerencia de Fronteras for their support and valuable insights.

## Author Contributions

**Conceptualization:** Gina Galindo, Daniel Romero, Daniel Rivera-Royero.

**Data curation:** Gina Galindo, Jhonattan Reales, Jhoan Castro, Daniel Romero, Sandra Rodriguez A.

**Formal analysis:** Gina Galindo, Jose Navarro, Jhonattan Reales, Jhoan Castro, Daniel Romero, Sandra Rodriguez A., Daniel Rivera-Royero.

**Funding acquisition:** Gina Galindo, Daniel Romero, Daniel Rivera-Royero.

**Investigation:** Gina Galindo, Jose Navarro, Jhonattan Reales, Jhoan Castro, Daniel Romero, Sandra Rodriguez A., Daniel Rivera-Royero.

**Methodology:** Gina Galindo, Jose Navarro, Jhonattan Reales, Jhoan Castro, Daniel Romero, Sandra Rodriguez A., Daniel Rivera-Royero.

**Project administration:** Gina Galindo.

**Resources:** Gina Galindo.

**Software:** Jose Navarro, Jhonattan Reales, Jhoan Castro.

**Supervision:** Gina Galindo, Daniel Romero, Sandra Rodriguez A., Daniel Rivera-Royero.

**Validation:** Gina Galindo, Jose Navarro, Jhonattan Reales, Jhoan Castro, Daniel Romero, Sandra Rodriguez A., Daniel Rivera-Royero.

**Visualization:** Gina Galindo, Jose Navarro, Jhonattan Reales, Jhoan Castro, Daniel Romero, Sandra Rodriguez A., Daniel Rivera-Royero.

**Writing – original draft:** Gina Galindo, Jose Navarro, Jhonattan Reales, Jhoan Castro, Daniel Romero, Sandra Rodriguez A., Daniel Rivera-Royero.

**Writing – review & editing:** Gina Galindo, Jose Navarro, Jhonattan Reales, Jhoan Castro, Daniel Romero, Sandra Rodriguez A.

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
