## [Decision Letter · Decision Letter 0]

5 Jan 2022

Immigrants resettlement in developing countries: a data-driven decision tool applied to the case of Venezuelan immigrants in Colombia

PONE-D-21-24079

Dear Dr. Galindo,

We’re pleased to inform you that your manuscript has been judged scientifically suitable for publication and will be formally accepted for publication once it meets all outstanding technical requirements.

Kind regards,

D. M. Basavarajaiah, ph.D

Academic Editor

PLOS ONE

Journal Requirements:

1. Please amend your Methods section to state where the databases used for the model can be accessed.

Reviewers' comments:

Reviewer's Responses to Questions

**Comments to the Author**

1. Is the manuscript technically sound, and do the data support the conclusions?

Reviewer #1: Yes

2. Has the statistical analysis been performed appropriately and rigorously? 

Reviewer #1: Yes

3. Have the authors made all data underlying the findings in their manuscript fully available?

Reviewer #1: Yes

4. Is the manuscript presented in an intelligible fashion and written in standard English?

Reviewer #1: Yes

5. Review Comments to the Author

Reviewer #1: In this manuscript, the authors propose a methodology to improve immigrants' socio-economic integration by adopting an algorithm developed in another research article.

1.The more detailed distinction between authors' work and adopted work.

2.It is recommended to add a proper description/ explanation to several figures. For example,

Fig 4: Explain k-means clustering method results presented using two principal axes.

Also, it is mentioned that Fig 4 numbered from left to right as clusters 1 to 4, however the legend given in the Fig 4 contradicts that statement.

Fig 6: Explain green, red, and gray limits in each histogram

3.The significance and limitation of this work should be discussed in more detail.

6. PLOS authors have the option to publish the peer review history of their article (what does this mean?). If published, this will include your full peer review and any attached files.

Reviewer #1: No

---

## [Editor Report · Acceptance letter]

13 Jan 2022

PONE-D-21-24079 

Immigrants resettlement in developing countries: a data-driven decision tool applied to the case of Venezuelan immigrants in Colombia 

Dear Dr. Galindo:

I'm pleased to inform you that your manuscript has been deemed suitable for publication in PLOS ONE. Congratulations! Your manuscript is now with our production department. 

Kind regards, 

on behalf of

Dr. D. M. Basavarajaiah 

Academic Editor

PLOS ONE